Corrected: Publisher correction

# Nanoplasmonic electron acceleration by attosecond-controlled forward rescattering in silver clusters

Johannes Passig[1], Sergey Zherebtsov[2,3], Robert Irsig[1], Mathias Arbeiter[1], Christian Peltz[1], Sebastian Göde[1], Slawomir Skruszewicz[1], Karl-Heinz Meiwes-Broer[1], Josef Tiggesbäumker[1], Matthias F. Kling[2,3] & Thomas Fennel [1,4]

In the strong-field photoemission from atoms, molecules, and surfaces, the fastest electrons emerge from tunneling and subsequent field-driven recollision, followed by elastic back-scattering. This rescattering picture is central to attosecond science and enables control of the electron's trajectory via the sub-cycle evolution of the laser electric field. Here we reveal a so far unexplored route for waveform-controlled electron acceleration emerging from forward rescattering in resonant plasmonic systems. We studied plasmon-enhanced photoemission from silver clusters and found that the directional acceleration can be controlled up to high kinetic energy with the relative phase of a two-color laser field. Our analysis reveals that the cluster's plasmonic near-field establishes a sub-cycle directional gate that enables the selective acceleration. The identified generic mechanism offers robust attosecond control of the electron acceleration at plasmonic nanostructures, opening perspectives for laser-based sources of attosecond electron pulses.

[1] Institut für Physik, Universität Rostock, Albert-Einstein-Str. 23, D-18059 Rostock, Germany. [2] Physik Department, Ludwig-Maximilians-Universität München, Am Coulombwall 1, D-85749 Garching, Germany. [3] Max-Planck Institut für Quantenoptik, Hans-Kopfermann-Straße 1, D-85748 Garching, Germany. [4] Max-Born-Institut für Nichtlineare Optik und Kurzzeitspektroskopie, Max-Born-Straße 2A, D-12489 Berlin, Germany. Correspondence and requests for materials should be addressed to J.Täu. (email: josef.tiggesbaeumker@uni-rostock.de) or to M.F.K. (email: matthias.kling@lmu.de) or to T.F. (email: thomas.fennel@uni-rostock.de)

When a bound electron is liberated by a strong laser field, it may escape or return to the residual potential to recombine or rescatter. This field-driven recollision is at the heart of strong-field science[1] and key to high-harmonic spectroscopy[2,3], laser-induced electron diffraction[4,5], and the generation of attosecond XUV pulses[6]. Moreover, the recollision concept provides a powerful framework for controlling the spectral[7,8], directional[9,10], and temporal properties[11] of the laser-driven electrons.

In atomic photoemission, i.e., for emission from nearly point-like potentials, elastic backscattering forms the rescattering plateau in the electron energy spectra with the well-known $10\,U_p$ cutoff (with $U_p$ the ponderomotive energy)[12], while forward rescattering only produces so-called low-energy structures[13–15]. These slow forward rescattered electrons enable electron holography through their interference with direct electrons[16]. Still, as for atoms and molecules, elastic backscattering also governs the near-field enhanced high-energy photoemission from the surface of nanotips, nanospheres, and plasmonic nanostructures[17–19]. For these extended systems, forward rescattering may even be quenched completely by electron scattering in the medium. As elastic backscattering dominates the fast electron emission from point-like targets and surfaces, similar protocols can be employed to achieve waveform control.

In a finite system like a cluster (Fig. 1a), however, forward rescattering is considered the major process for high-energy electron emission[20,21]. The underlying concept, being also the key trick in laser grating accelerators[22], is that the electron's travel distance per optical cycle can match the scale of the system's static or induced potential. In consequence, optical waveform control of scale-matched forward rescattering should be possible and can be studied using clusters as model systems. The ramifications of such control would be of general interest for electron acceleration based on plasmonic nanostructures.

The ability of clusters to efficiently absorb intense laser light[23] facilitates the formation of nanometer-sized plasmas that emit X-rays[24], energetic[25], and highly charged ions[26], as well as fast electrons. The emerging deep cluster potential strongly confines plasma electrons and permits robust plasmonic resonance excitation after appropriate cluster expansion[27]. Besides high absorption, resonance excitation also triggers particularly energetic electron emission along the polarization axis that evidences field-driven acceleration[28]. According to theory, the acceleration proceeds in a single laser cycle during the electron's final transit through the cluster, reflecting the above mentioned forward rescattering with matched scales. Both direct laser field-driven acceleration[20] and acceleration by the cluster polarization field have been put forward to explain the enhanced energy gain[21]. In

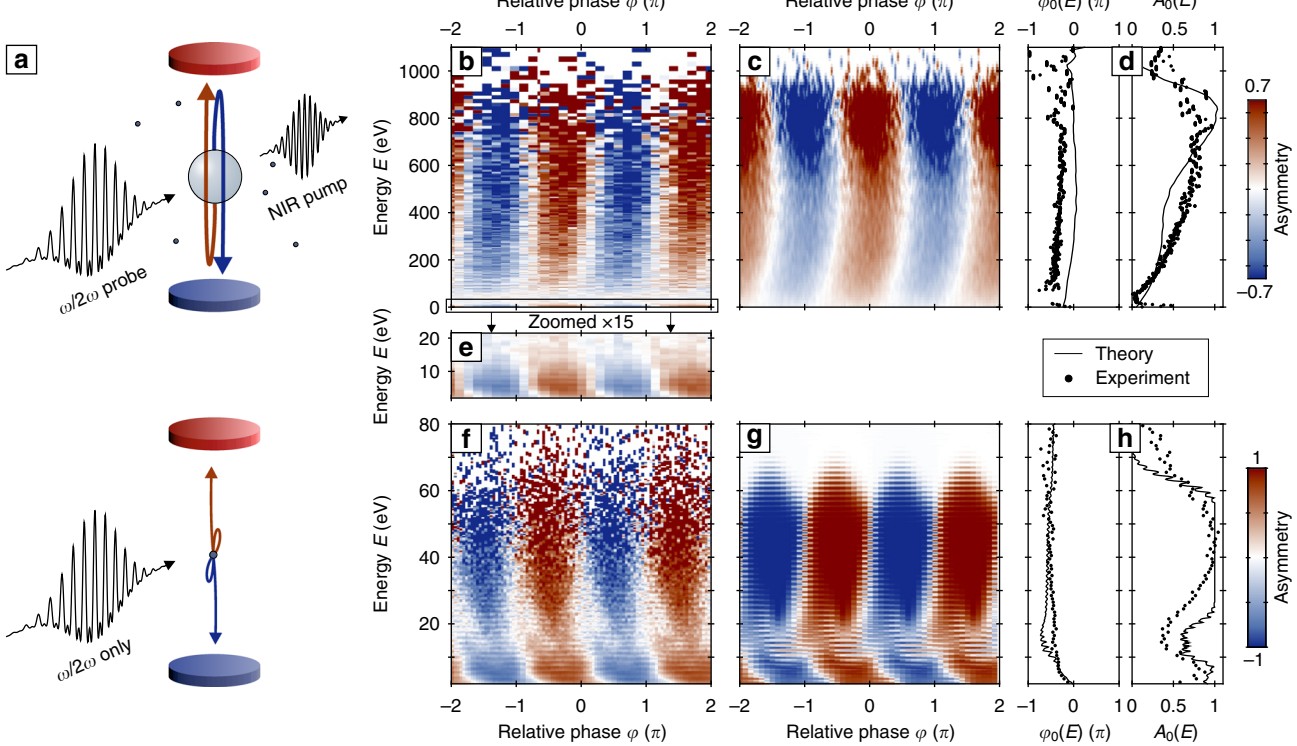

**Fig. 1** Phase-dependent high-energy electron emission. **a** Schematics of the experiment and dominant rescattering processes in the electron emission from clusters (top) and atoms (bottom). Dots in the upper part indicate residual gas atoms present in the cluster experiment. Blue and red discs indicate the electron detectors. **b** Energy-dependent emission asymmetry $A(E, \varphi) = \frac{Y_{up} - Y_{down}}{Y_{up} + Y_{down}}$ of the electron spectra $Y_{up}(E)$ and $Y_{down}(E)$ measured with the upward and downward detector, respectively. The data show the asymmetry evolution with relative phase $\varphi$ for two-color excitation of pre-expanded silver clusters with $\overline{R} \approx 3\,nm$ initial radius. **c** Corresponding molecular dynamics (MD) simulation result for $R = 3\,nm$ clusters (Ag$_{7500}$). **d** Asymmetry amplitudes $A_0$ and asymmetry phases $\varphi_0$ from harmonic fits of the data in **b**, **c** with $A_{fit}(E, \varphi) = A_0(E)\cos(\varphi - \varphi_0(E))$. **e** Zoom into the low-energy atomic contribution from residual Ar gas in the measured cluster data. **f** Measured asymmetry from pure atomic Ar gas (two-color pulse only). Note the different energy scales in the upper and lower panels. **g** Asymmetry for atomic Ar as predicted by the numerical solution of the time-dependent Schrödinger equation (TDSE) in single active electron approximation. Matching TDSE results to experiment yields the spectral intensities $I_\omega = 4 \times 10^{13}\,W\,cm^{-2}$ and $I_{2\omega}/I_\omega = 0.2$ of the bichromatic probe pulse and calibrates the phase axis. **h** Fit results for the atomic data; The upper color bar for the asymmetry corresponds to the maps shown in **b**, **c**, **e**, while the lower color bar corresponds to **f**, **g**. A 120 fs NIR pump pulse (800 nm) at intensity $I_{pump} = 10^{14}\,W\,cm^{-2}$ arriving 1.4 ps before the two-color probe pulse is used to pre-expand clusters

both cases, the forward rescattering process should be sensitive to the field evolution in the relevant laser cycle. An experimental demonstration of this dependence, however, was still lacking.

In the current study we observe the waveform-dependent electron emission from resonantly driven silver clusters. We find directional control with the phase of a bichromatic laser field,

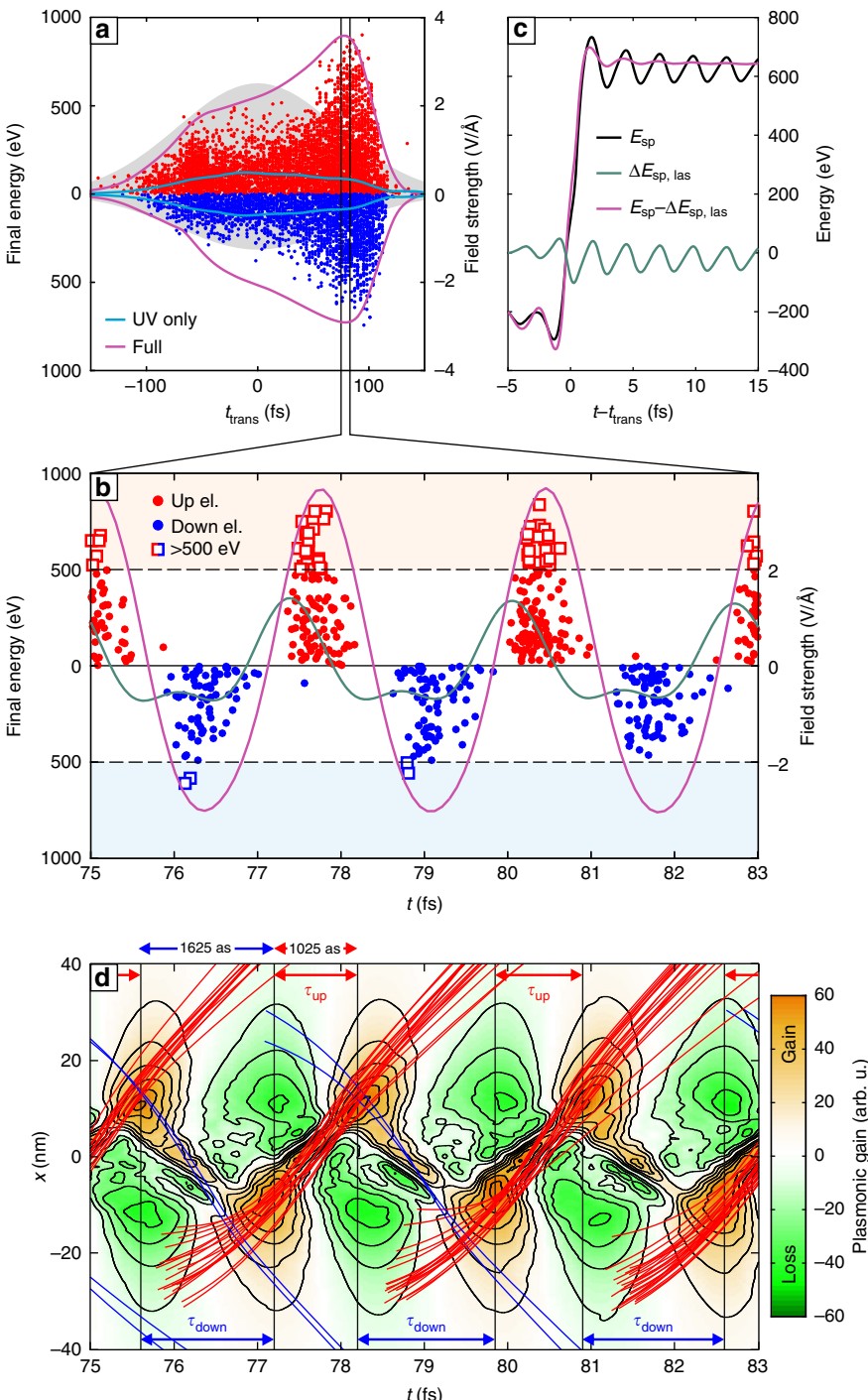

**Fig. 2** Molecular dynamics simulation of the asymmetric electron emission. **a** Scatter plot of final kinetic energies and emission times ($E_{final}$, $t_{trans}$) of electrons emitted in upward (red) and downward (blue) direction from the simulation run for $\varphi = 0$ in Fig. 1c. The envelopes of the laser field and the indicated polarization field components inside the cluster are shown as gray area and solid curves, respectively (only components along $x$-axis, right scale). The time axis is centered at the peak of the $\omega/2\omega$ probe pulse. **b** Cycle-resolved analysis of the electron emission times and energies (symbols, left scale) in the resonance region. Solid curves show the corresponding evolutions of the laser field (green) and the polarization field (magenta) inside the cluster (right scale). Fast electrons ($E_{final} > 500$ eV, marked with boxes) are selected for further analysis. **c** Evolution of the single-particle energy $E_{sp}$ during the final transit (black) resulting from averaging over the selected trajectories. Subtracting the direct laser absorption $\Delta E_{sp,las} = q_i \int \mathbf{r}_i \cdot \mathcal{E}_{las} dt$ (green) yields the plasmonic contribution (magenta). **d** Projected trajectories $x_i(t)$ of the selected electrons and spatiotemporal plasmonic gain map $\frac{\partial}{\partial t} V(x, t)$ sampled on the polarization axis. Dark orange regions mark high-gain points and horizontal arrows indicate optimal gate times for forward rescattering in upward and downward direction, respectively. At the time of the resonance, the cluster has expanded to a radius $R \approx 13$ nm

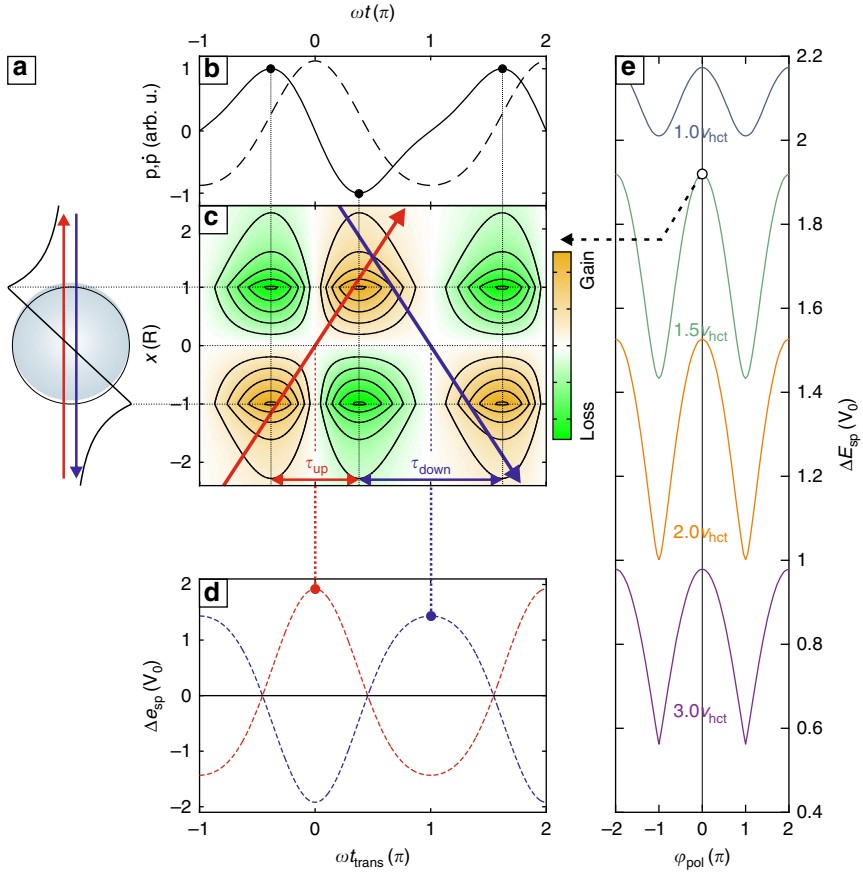

**Fig. 3** Two-color nanoaccelerator model. **a** Schematics of scale-matched forward rescattering at a polarized sphere with potential $V(x, t) = V_{pol}(x)p(t)$, where $V_{pol}(x)$ is the static polarization potential (black curve) for unit dipole moment with amplitude $V_0$. **b** Evolution of the two-color dipole moment $p(t)$ (dashed curve) for a relative amplitude $\gamma_{pol} = 1/8$ and corresponding dipole acceleration $\dot{p}(t)$ (solid curve). **c** Resulting gain map $\frac{\partial}{\partial t} V = V_{pol}(x)\dot{p}(t)$ with pronounced high-gain points (orange). Electron drift trajectories $x(t) = (t - t_{trans})v_d$ for optimal passages in up (red) and down (blue) direction for drift velocity $v_d = 1.5v_{hct}$ and relative dipole phase $\varphi_{pol} = 0$. Corresponding transit times $t_{trans}$ are indicated as vertical dotted lines. Here $v_{hct}$ is the half cycle transit velocity (see text). **d** Transit time-dependent gain in the perturbative limit $\Delta E_{sp}(t_{trans}) = \int \frac{\partial}{\partial t} V_{pol}(x(t), t) dt$ for upward and downward electrons. The respective peak values (circular symbols) define the optimal trajectories. **e** Optimal gain for upward electrons vs. relative dipole phase $\varphi_{pol}$ for selected drift velocities $v_d$

identify the underlying near-field mediated gating mechanism, and describe its implications for the control of attosecond electron pulses.

## Results

**Phase-dependent directional electron emission.** In our experiment, silver clusters are first pre-excited by an intense near-infrared (NIR) pump pulse (Fig. 1a; Methods). The pump step alone yields only weak electron emission and low energies. It triggers cluster expansion, establishing a plasmonic resonance with the fundamental wave within about 1–2 ps. Subsequent waveform-controlled excitation was utilized with a delayed $\omega/2\omega$ probe pulse (Methods), containing a dominant NIR fundamental and a weaker second harmonic component ($\gamma^2 = I_{2\omega}/I_\omega \approx 0.2$). The sub-cycle waveform of the linearly polarized $\omega/2\omega$ probe field $\mathcal{E}_{las}(t) = \mathcal{E}_0 \mathbf{e}_x[\cos(\omega t) + \gamma \cos(2\omega t + \varphi)]$ is controlled with the relative phase $\varphi$. Direction-resolved energy spectra are recorded along the polarization axis with a stereo electron spectrometer (Methods). For resonant $\omega/2\omega$-probe excitation, we find strongly enhanced and highly phase-dependent directional acceleration, characterized by up–down emission asymmetries $A(E,\varphi)$ up to $\pm 0.7$ (Fig. 1b–d). The retrieved asymmetry phase $\varphi_0(E)$ denotes the value of the relative phase $\varphi$ with maximal upward emission and is spectrally nearly flat (Fig. 1d). The observed phase

dependence evidences robust waveform control up to keV kinetic energies (here beyond 300 $U_p$), exceeding the energies found for $\omega/2\omega$ excitation of Ar carrier gas (probe pulse only) by more than an order of magnitude (Fig. 1f). The atomic asymmetry features—bearing fascinating physics in itself[29]—remain clearly visible in the cluster spectra (Fig. 1e) and enable calibration of laser parameters and the phase axis through comparison with quantum simulations based on the solution of the time-dependent Schrödinger equation (TDSE) (Fig. 1g, h).

**Simulation of the directional electron acceleration.** The phase-controlled acceleration is analyzed with semiclassical molecular dynamics (MD) simulations (Methods). Up to a small offset in the asymmetry phase (Fig. 1d) that is discussed below, the predicted electron energies and asymmetry features show good agreement with experiment (Fig. 1b–d), supporting that the directional acceleration process is well-captured. We note that the most energetic electrons in the experiment are expected to stem from the larger clusters in the size distribution. Simulations with various cluster sizes confirm this trend.

A representative MD run with relative phase $\varphi = 0$, where the asymmetry becomes most apparent, reveals the main physics (Fig. 2). For each emitted electron trajectory, the moment of escape is characterized by the transit time, $t_{trans}$, associated with

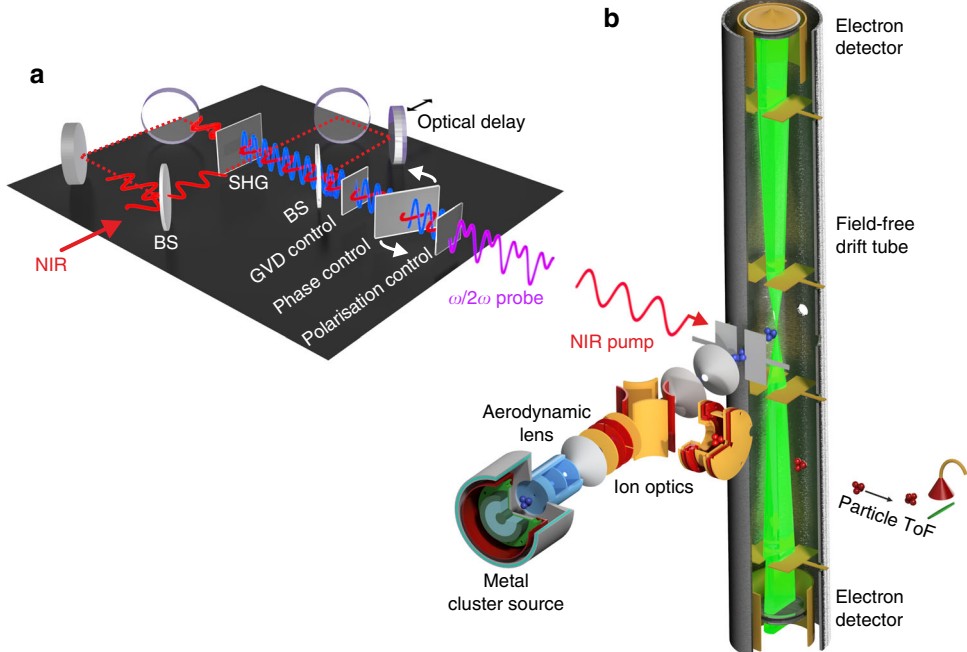

**Fig. 4** Detailed experimental setup. **a** Mach–Zehnder type interferometer to generate the near-infrared (NIR) pump pulse and the delayed, phase-controlled bichromatic ($\omega/2\omega$) probe pulse. The incoming NIR pulse is split at the first beam splitter (BS). A beta-barium-borate (BBO) crystal placed in one of the arms is utilized for second harmonic generation (SHG) prior to the recombination of the two beams at the second beam splitter. A set of birefringent crystals is used for compensating the group velocity dispersion (GVD) and for controlling the relative phase. The polarization of the NIR and SHG components is turned parallel with a wave plate prior to the focussing of the beam. **b** Cluster apparatus consisting of the magnetron sputtering cluster source followed by an aerodynamic lens system. The cluster size distribution is monitored using electrostatic deflection into a particle time-of-flight spectrometer (particle TOF). Without deflection, the cluster beam enters a field-free stereo electron time-of-flight spectrometer through a set of skimmers, where photoemitted electrons are detected

the final transit through the cluster via $x(t_{\mathrm{trans}}) = 0$. The most energetic electrons escape in a short period of resonant coupling in the trailing edge of the $\omega/2\omega$ pulse (Fig. 2a), underlining the transient plasmonic resonance excitation. The strong resonant enhancement can be clearly assigned to the NIR contribution of the polarization field, while the second harmonic contribution follows the pulse envelope (cf. violet and blue curves in Fig. 2a). The attosecond bursts of fast electrons emitted in each half cycle at resonance are strongly correlated with the sub-cycle structure of the induced polarization field inside the cluster regarding emission time, direction, and energy (Fig. 2b). Note that the fundamental and second harmonic contributions of the polarization field have a nearly vanishing relative phase in the period of the high-energy emission, as discussed in more detail later. The evolution of the single-particle energy

$$E_{\mathrm{sp}} = \frac{m}{2}\left|\dot{\mathbf{r}}_i\right|^2 + V(\mathbf{r}_i(t), t) \qquad (1)$$

for corresponding trajectories $\mathbf{r}_i(t)$, where $i$ is the particle index, reveals energy gains approaching 1 keV in a single fundamental laser cycle (Fig. 2c). Here $V(\mathbf{r}, t)$ is the potential created by all other charges. The associated single-particle energy gain rate

$$\dot{E}_{\mathrm{sp}} = q_i \dot{\mathbf{r}}_i(t) \cdot \mathcal{E}_{\mathrm{las}} + \frac{\partial}{\partial t} V(\mathbf{r}_i(t), t) \qquad (2)$$

contains the direct gain from the laser field and the plasmonic contribution due to interaction with the dynamical polarization (Methods). A selective gain analysis reveals an essentially negligible net gain from the laser field for fast electrons (Fig. 2c), such that their asymmetric acceleration must stem from the plasmonic term.

**Sub-cycle gating of the acceleration process**. The arrangement of fast trajectories in the spatiotemporal plasmonic gain map $\frac{\partial}{\partial t} V(\mathbf{r}, t)$ in Fig. 2d uncovers the underlying mechanism. Electrons emitted upward or downward undergo single-cycle forward rescattering and are clearly aligned to two pronounced high-gain regions (orange areas)—the first when entering and the second when exiting the cluster. The strong concentration of all trajectories near these high-gain points emphasises their role as a spatiotemporal gate for the acceleration process.

Key to explaining the up–down asymmetry are the different gate times $\tau_{\mathrm{up}}$ and $\tau_{\mathrm{down}}$. The gate times reflect the delay between the high-gain points associated with corresponding transits (Fig. 2d). The direction-dependent gate times directly reflect the two-color polarization induced by the bichromatic laser field. Most importantly, since shorter gate times support faster transits, the acceleration can be enhanced for one direction, here for upward electrons (red lines in Fig. 2). The fastest downward electrons (blue lines in Fig. 2d) pass the cluster too quickly and miss the second high-gain point.

**Universal two-color nanoaccelerator model**. The universal nature of the phase-controlled acceleration and its implications for forming or modifying attosecond electron bunches is highlighted by a one-dimensional nanoaccelerator model. Therein, electrons are considered to cross a dynamically polarized sphere (Fig. 3a) with potential $V(x, t) = V_{\mathrm{pol}}(x) p(t)$. Here $p(t) = P_0[\cos(\omega t) + \gamma_{\mathrm{pol}} \cos(2\omega t + \varphi_{\mathrm{pol}})]$ is the two-color dipole moment (Fig. 3b) with relative dipole phase $\varphi_{\mathrm{pol}}$ and relative amplitude $\gamma_{\mathrm{pol}}$ of the $2\omega$ component and $V_{\mathrm{pol}}(x)$ is the static polarization potential. The laser field creating the polarization is neglected.

Similar to MD, the plasmonic gain rate map $\frac{\partial}{\partial t} V(x, t) = V_{\mathrm{pol}}(x) \dot{p}(t)$ displays pronounced high-gain regions. Maxima are

located at the sphere surface and occur at the peaks of the dipole velocity $\dot{p}$, yielding asymmetric gates for upward and downward trajectories (Fig. 3c). In the limit of small dipole amplitudes $P_0$, the energy gain can be approximated using unperturbed drift trajectories $x(t) = (t - t_{trans})v_d$ with drift velocity $v_d$. Focussing on the timing, the energy gain is maximal for transits centered in the respective gate (Fig. 3d), see optimal trajectories in Fig. 3c. The positive (negative) first derivative of the energy gain for earlier (later) transits would induce a spectral upchirp (downchirp) of a passing ultrashort electron bunch, with a phase-dependent chirp profile. The maximal gain for upward electrons shows a robust phase dependence up to high drift velocities and peaks at relative dipole phase $\varphi_{pol} = 0$ for all electrons with $v_d \geq v_{hct}$ (Fig. 3e). Here $v_{hct} = 2R\omega/\pi$ is the half cycle transit velocity, which also defines the characteristic energy scaling of the forward acceleration process with cluster size and laser frequency $E_{hct} \propto R^2\omega^2$. Manipulation of the gate times via the relative dipole phase thus establishes a systematic route to optically control the direction specific acceleration and chirp of attosecond electron pulses.

For comparison with experiment, relative laser and dipole phases, $\varphi$ and $\varphi_{pol}$, have to be connected. A damped harmonic oscillator response with spectral phase lags $\Delta\varphi_\omega$ and $\Delta\varphi_{2\omega}$ yields $\varphi_{pol} = 2\Delta\varphi_\omega - \Delta\varphi_{2\omega} + \varphi$. For resonance excitation by the fundamental and weak damping ($\Delta\varphi_\omega = \pi/2$ and $\Delta\varphi_{2\omega} = \pi$) the laser and dipole relative phases are equal such that the requirement for maximal asymmetry, $\varphi_{pol} = 0$, is met for $\varphi = 0$. Deviations from this value in the experimental data and MD results (Fig. 1b, c) is ascribed to a modified phase lag, e.g., $\Delta\varphi_{2\omega} < \pi$, due to plasmon damping. Smaller shifts in MD are attributed to incomplete account of electron–ion collisions. Most importantly, the system-specific spectral phase lag plays a key role in the acceleration and may even be used as a control parameter in the multi-stage acceleration process in nanostructure arrays.

## Discussion

In summary, control of resonant plasmon-assisted forward rescattering via the waveform of a bichromatic laser field is observed for clusters and found to rely on a universal gating mechanism. The process is generic for finite systems and governed by the polarization field and thus effectively decoupled from the laser field. The waveform-controlled forward acceleration is expected to enable unprecedented spectral, temporal, and directional compression of sub-cycle electron bunches in optical accelerators, e.g., using cascaded hollow plasmonic structures as nano-resonators. This opens scientific and technological perspectives for low-cost high-quality sources of ultrashort electron pulses for applications in electron microscopy and spectroscopy[22,30]. The forward acceleration scheme further motivates new acceleration modes in thin foils. Waveform-controlled superintense pulses, as used for controlled high-harmonic generation[31], could induce the strong longitudinal polarization to drive forward rescattering up to the relativistic regime.

## Methods

**Experimental setup**. Silver clusters are generated by gas-mediated aggregation of silver vapor from magnetron sputtering[32] and delivered to the interaction region as a narrow molecular beam with Ar as a carrier gas (Fig. 4). Aerodynamic lenses[33] and a dual pumping stage are used to maintain sufficient target density and low pressure in the interaction region. The cluster beam is deflected with ion optics into a mass spectrometer for size characterization. The log-normal size distribution of the clusters peaks at 4 nm diameter, corresponding to a mean diameter of ca. 6 nm ($N \approx 7000$) as confirmed by particle mass spectrometry[34] and TEM analyses. The cluster beam has a diameter of 2 mm when entering the interaction region. The experiments are performed in the limit, where only a single cluster is excited in each laser shot. The stereo time-of-flight (TOF) spectrometer contains two opposite, field-free and magnetically shielded 0.5 m drift tubes (Fig. 4b) and accepts electrons in a solid angle of 0.07 Sr for detection by a digitizer system (Acqiris DC271) operating in single-event counting mode.

NIR laser pulses (800 nm, 120 fs) from a 1 kHz laser system (Solstice, Spectra Physics) are split into pump and probe pulses using a Mach–Zehnder interferometer (Fig. 4a). The 400 nm component in the probe arm is generated via second harmonic generation in a beta-barium-borate (BBO) crystal. After recombination, the relative phase of the two spectral components is adjusted with a rotatable birefringent plate. A wave plate (zero order for 800 nm) is used to rotate the polarizations such that all beams are polarized along the axes of the TOF spectrometer (Fig. 4b). The collinear pump-probe beam is focussed by a spherical mirror ($f = 0.5$ m) into the interaction region, where it intersects the molecular beam. The laser focus is placed 3.5 mm behind the molecular beam to reduce phase averaging due to the Gouy phase shift.

**MD simulation method**. The waveform-dependent electron acceleration in the clusters is calculated by hybrid MD simulations that employ a microscopic classical treatment of the electron and ion dynamics in the laser-driven nanoplasma[35]. Atomic tunneling and electron impact ionization events are treated via effective rates. The classical propagation of plasma particles uses a regularized Coulomb interaction that is matched to atomic ionization energies and impedes classical electron–ion recombination below the atomic energy levels. The force evaluation in the gridless code is accelerated by exploiting massively parallel processing. Local field effects such as field enhancement and the depression of ionization potentials due to plasma screening are included in the evaluation of tunneling and electron impact ionization, respectively. Spherical silver clusters are initialized with fcc structure and a single active conduction electron. The conduction electrons were initialized with 3 eV kinetic energy, which roughly reflects the average valence electron kinetic energy in the bulk given by $3/5E_F$, where $E_F = 5.5$ eV is the Fermi energy of silver. The simulations cover the full dynamics resulting from the excitation with pump and probe pulses. For the considered cluster sizes, the laser field can be described in dipole approximation. Electromagnetic effects are neglected. Within the resonant pump-probe excitation on average 8.2 additional electrons were removed from each atom, leading to a total inner ionization of $q_{ii} = 9.2$ per atom. The electron emission into the continuum almost exclusively takes place in the probe pulse, reaching a total outer ionization degree of $q_{oi} = 3.1$ for the considered parameters.

**Selective energy gain analysis**. Within the classical propagation, particle trajectories are integrated according to Newton's equation of motion

$$m_i\ddot{\mathbf{r}}_i = q_i\mathcal{E}_{las}(t) - \nabla_{\mathbf{r}_i}V(\mathbf{r}_i, t), \qquad (3)$$

where $m_i$ and $q_i$ are the mass and charge of the respective plasma particle, $\mathcal{E}_{las}(t)$ is the laser field, and $V(\mathbf{r}_i, t)$ is the instantaneous Coulomb potential created by all other charges at the position $\mathbf{r}_i$ of the considered actual particle. Further, from Eq. (1) the rate of change of the single-particle energy follows as

$$\frac{d}{dt}E_{sp} = m_i\dot{\mathbf{r}}_i \cdot \ddot{\mathbf{r}}_i + [\nabla_{\mathbf{r}_i}V(\mathbf{r}_i, t)] \cdot \dot{\mathbf{r}}_i + \frac{\partial}{\partial t}V(\mathbf{r}_i, t). \qquad (4)$$

Inserting the equation of motion Eq. (3) in the first term on the right hand side of Eq. (4) immediately yields the splitting of single-particle energy change rate in Eq. (2) into the purely laser-driven component and the plasmonic component mediated by the dynamical evolution of the system's space-charge and polarization field.

**Data availability**. All measured data used in this study as well as the custom parts of the simulation code are available on request from the corresponding authors.

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

## Acknowledgements

We acknowledge financial support by the Deutsche Forschungsgemeinschaft via SFB 652, SPP 1840, and a Heisenberg Fellowship (FE 1120/4). S.Z. and M.F.K. acknowledge support from the EU via the ERC grant ATTOCO (307203) and from the DFG via the excellence cluster "Munich Centre for Advanced Photonics". Computer time has been provided by the North German Supercomputing Alliance (HLRN) within project mvp00010.

## Author contributions

J.P., J.T., K.-H.M.-B., M.F.K., and T.F. conceived the experiment. J.P., R.I., S.Z., S.G., S.S., and T.F. performed the measurements. T.F. and C.P. developed the MD simulations and performed the theoretical analysis. M.A. performed TDSE calculations. J.P., M.F.K., J.T., K.-H.M.-B., C.P., and T.F. evaluated, analyzed, and interpreted the results. T.F. wrote the manuscript with the input from all authors.

## Additional information

**Competing interests:** The authors declare no competing financial interests.

