## [Peer Review File · Nature Communications]

Reviewers' comments:

Reviewer #1 (Remarks to the Author):

In their extensive experimental and theoretical work, Passig et al. report on a novel scheme to accelerate electrons by forward scattering mediated by the field inside an expanding nanoplasmonic particle. In their experiment, electron energies up to about 1keV are readily achieved from Ar nanoclusters. The acceleration process can be controlled by using a two-color field consisting of a strong IR and its second harmonic; tuning the relative phase of the two colors determines the amount and the direction of the acceleration process. Two theory models reveal the acceleration mechanism: An electron which crosses the expanding nanoparticle in a forward scattering process is accelerated strongly by the internal polarization field when its trajectory is well timed. The time window of acceleration can be tuned with two-color field. The authors' findings might lead to a new technique for linear, laser-based electron accelerators.

In my opinion, the manuscript is well-written and contains all relevant information. The experimental and theoretical methods and results and their interpretation are of high quality and described in great clarity. The citation of prior work is adequate. I believe that the work definitely deserves publication in Nature Communications. I only have minor comments to the paper (in descending order of importance):

- p. 3, Fig. 2b: The polarization field is supposed to resemble the two-color field to some extent (cf. p.4, $\phi_{\text{pol}} = 0$ if $\phi = 0$). In the figure showing the polarization according to the MD simulation, the asymmetric features seen in the waveform of the two-color field are smoothed out. Is this the result of a different SH-IR mixing ratio in the induced polarization compared to the two-color field? Or is some smoothing or averaging process involved?

- p.2/3, sentence "A representative MD run": In order to assist the reader better, I recommend adding a more detailed comment that you focus on the representative case $\phi = 0$ where the asymmetry effect becomes most apparent.

- p.1, first paragraph of Main Text: Forward rescattering does not only lead to low-energy structures, but also enables electron holography through the interference of direct and forward scattered electrons (see e.g. Huisman et al., Science 2011). I believe that this application should be mentioned.

After properly addressing my (minor) comments, for which I fully trust the authors, I fully recommend publication in Nature Communications.

Reviewer #2 (Remarks to the Author):

A) What are the major claims of the paper?

The manuscript claims to show control over the spatio-temporal emission of almost keV electrons by controlling the sub-cycle waveform of a two-color light field. Matching the result of an experiment with two-levels of theory (quantum and hybrid quantum/classical) the authors present a compelling explanation the high-energy electron emission. The explanation leads to the other major claim: the mechanism is generic and can thus be applied to other systems to create sub-cycle high energy electron bursts.

All the claims are well substantiated by the authors via the evidence presented. Their experimental setup was well designed for this experiment and there is little doubt of the validity of those results. The quantum and hybrid modeling were also well done and appropriately applied to this system and thus the matching of the experimental and theory results is sufficient evidence for the manuscripts major claims.

The final section of the manuscript verifies the findings with a simplified model of the process. This is well thought out and demonstrates the underlying physics behind the acceleration mechanism.

B) Are they novel and will they be of interest to others in the community and the wider field?

The mechanism is shown to be of a generic nature and is thus of interest to a broad audience including but not limited to: the laser physics community, plasmonic researchers and the electron microscopy and spectroscopy communities. By controlling the waveform, something the community is now very good at, the two-color pulse can be used to adjust the timing and direction of the electron emission. This opens up many possibilities including that of multistage acceleration using a nanostructure array, for instance.

C) Is the work convincing, and if not, what further evidence would be required to strengthen the conclusions?

The work is very convincing and in addition provided excellent insight to the mechanism.

D) On a more subjective note, do you feel that the paper will influence thinking in the field? Please feel free to raise any further questions and concerns about the paper.

The paper will very likely spark additional work both tailoring and optimizing this result as well as using for other research areas. The result is clearly demonstrated and generic enough to be of interest to many different groups

The following are questions which were raised during my review. I do not feel they are required to be addressed in the manuscript for publication as a condition. However, their answers may address other reader's similar questions about the mechanism.

1) Is there any indication that the highest energy electrons are coming from the larger clusters? The experimental setup allows for single laser-cluster per shot. Thus the data is not averaged on a per shot basis. The MD results and nanoaccelerator model suggest that the acceleration is from a single pass through the cluster and thus the question arises as to will a larger cluster give more acceleration? Or is there a limit based on the mean-free-path of the electrons and thus it is actually the smaller clusters in the cluster size distribution that are causing the highest energy acceleration?

2) What was the initial state of the MD-silver clusters?

The manuscript states the system was initialized with all Ag as +1 in an fcc. Were the electrons initialized with a total energy equal to their ionization potential or some global measure of the electron plasma temperature? Additionally, was the system simulated for both the pump probe or just the probe?

3) What was the average final charge state of the silver clusters after irradiation? The main reason for this is to get an idea for the total number of electrons released with high energy and what proportion that represents to the total number of ionized electrons.

4) Was the full spatial extent of the laser pulse's used or where the pulses "clipped", removing the lower intensity wings of the pulse? This would help to explain if the low intensity signal is noise and it is just the high intensity region that provides the fast electrons or if it is some region of intensity that has the optimal parameters.

Typos

The paper is well written, but there are multiple mistakes of s → z that are incorrectly spelled: ionization, characterized and polarization.

D) We would also be grateful if you could comment on the appropriateness and validity of any statistical analysis, as well the ability of a researcher to reproduce the work, given the level of detail provided.

The statistical techniques used are appropriate as are the numerical techniques used were also appropriate. The manuscript does provide sufficient details which will allow for others to reproduce the results and build on them.

This review was conducted by: Edward Ackad

Response to the reviewer #1

(comments of reviewer #1 are in **blue color** and our response is shown in **black**)

Reviewer #1: In their extensive experimental and theoretical work, Passig et al. report on a novel scheme to accelerate electrons by forward scattering mediated by the field inside an expanding nanoplasmonic particle. In their experiment, electron energies up to about 1keV are readily achieved from Ar nanoclusters. The acceleration process can be controlled by using a two-color field consisting of a strong IR and its second harmonic; tuning the relative phase of the two colors determines the amount and the direction of the acceleration process. Two theory models reveal the acceleration mechanism: An electron which crosses the expanding nanoparticle in a forward scattering process is accelerated strongly by the internal polarization field when its trajectory is well timed. The time window of acceleration can be tuned with two-color field. The authors' findings might lead to a new technique for linear, laser-based electron accelerators.

In my opinion, the manuscript is well-written and contains all relevant information. The experimental and theoretical methods and results and their interpretation are of high quality and described in great clarity. The citation of prior work is adequate. I believe that the work definitely deserves publication in Nature Communications.

We thank the reviewer for the supportive statements.

Reviewer #1: I only have minor comments to the paper (in descending order of importance):

- p. 3, Fig. 2b: The polarization field is supposed to resemble the two-color field to some extent (cf. p.4, $\phi_{\text{pol}} = 0$ if $\phi = 0$). In the figure showing the polarization according to the MD simulation, the asymmetric features seen in the waveform of the two-color field are smoothed out. Is this the result of a different SH-IR mixing ratio in the induced polarization compared to the two-color field? Or is some smoothing or averaging process involved?

The reviewer is fully correct - it is the relative SH contribution in the polarization field that is lower than in the laser field, because the field enhancement of the SH is smaller than for the fundamental. The stronger IR enhancement directly reflects the resonance situation with respect to the IR field component. As a result the asymmetry is reduced in the polarization field. In fact, the relative phase of the polarization field is close to zero in the period of most energetic electron emission (i.e. almost equal to the relative phase of the laser), fully supporting the conclusion from the simpler nanoaccelerator model. We agree that these points should be spelled out explicitly and made the following amendments.

changes to the manuscript:

- we added the envelope of the SH contribution of the polarization field in Fig 2a to show that the IR polarization is responsible for the resonance enhancement and updated the figure caption.
- We implemented the discussion of the spectral polarization field ratio in the text of the corresponding paragraph: *“The most energetic electrons escape in a short period of resonant coupling in the trailing edge of the $\omega/2\omega$ pulse (Fig. 2a), underlining the transient plasmonic resonance excitation. The strong resonance enhancement can be clearly assigned to the NIR contribution of the polarization field while the second harmonic contribution follows the pulse envelope (cf. purple and blue curves in Fig. 2. The attosecond bursts of fast electrons emitted in each half-cycle at resonance are strongly correlated with the sub-cycle structure of the induced polarization field inside the cluster regarding emission time, direction, and energy (Fig. 2b). Note that the fundamental and second harmonic contributions of the polarization field have a*

nearly vanishing relative phase in the period of the high energy emission, as discussed in more detail later.”

- p.2/3, sentence “A representative MD run”: In order to assist the reader better, I recommend adding a more detailed comment that you focus on the representative case $\phi = 0$ where the asymmetry effect becomes most apparent.

Yes, the choice for this run was made because it shows a strong asymmetry and is therefore ideal to trace the underlying mechanism.

changes to the manuscript:

- As suggested we added the statement suggested by the reviewer in the text:
“A representative MD run with relative phase $\phi=0$ where the asymmetry becomes most apparent reveals the main physics (Fig. 2).”

- p.1, first paragraph of Main Text: Forward rescattering does not only lead to low-energy structures, but also enables electron holography through the interference of direct and forward scattered electrons (see e.g. Huisman et al., Science 2011). I believe that this application should be mentioned.

We thank the reviewer for pointing out this important implication of forward rescattering and added the suggested reference to the text.

changes to the manuscript:

- suggested reference added and implemented in the revised text :
“These slow forward rescattered electrons enable electron holography through their interference with direct electrons [HuismanS2011]. Still, as for the fast electrons from atoms and molecules, elastic backscattering also governs the near-field enhanced high-energy photoemission from nanotips, nanospheres, and plasmonic nanostructures [17,18,19], for which forward rescattering may even be quenched completely by electron scattering in the medium.”

After properly addressing my (minor) comments, for which I fully trust the authors, I fully recommend publication in Nature Communications.

We hope that all comments have been addressed satisfactorily.

Response to the reviewer #2

(comments of reviewer #2 are in **green color** and our response is shown in **black**)

Reviewer #2:

A) What are the major claims of the paper?

The manuscript claims to show control over the spatio-temporal emission of almost keV electrons by controlling the sub-cycle waveform of a two-color light field. Matching the result of an experiment with two-levels of theory (quantum and hybrid quantum/classical) the authors present a compelling explanation the high-energy electron emission. The explanation

leads to the other major claim: the mechanism is generic and can thus be applied to other systems to create sub-cycle high energy electron bursts.

All the claims are well substantiated by the authors via the evidence presented. Their experimental setup was well designed for this experiment and there is little doubt of the validity of those results. The quantum and hybrid modeling were also well done and appropriately applied to this system and thus the matching of the experimental and theory results is sufficient evidence for the manuscripts major claims. The final section of the manuscript verifies the findings with a simplified model of the process. This is well thought out and demonstrates the underlying physics behind the acceleration mechanism.

B) Are they novel and will they be of interest to others in the community and the wider field?

The mechanism is shown to be of a generic nature and is thus of interest to a broad audience including but not limited to: the laser physics community, plasmonic researchers and the electron microscopy and spectroscopy communities. By controlling the waveform, something the community is now very good at, the two-color pulse can be used to adjust the timing and direction of the electron emission. This opens up many possibilities including that of multistage acceleration using a nanostructure array, for instance.

C) Is the work convincing, and if not, what further evidence would be required to strengthen the conclusions?

The work is very convincing and in addition provided excellent insight to the mechanism.

D) On a more subjective note, do you feel that the paper will influence thinking in the field? Please feel free to raise any further questions and concerns about the paper.

The paper will very likely spark additional work both tailoring and optimizing this result as well as using for other research areas. The result is clearly demonstrated and generic enough to be of interest to many different groups

We are grateful for the positive and elaborate evaluation of our work.

Reviewer #2: The following are questions which were raised during my review. I do not feel they are required to be addressed in the manuscript for publication as a condition. However, their answers may address other reader's similar questions about the mechanism.

1) Is there any indication that the highest energy electrons are coming from the larger clusters? The experimental setup allows for single laser-cluster per shot. Thus the data is not averaged on a per shot basis. The MD results and nanoaccelerator model suggest that the acceleration is from a single pass through the cluster and thus the question arises as to will a larger cluster give more acceleration? Or is there a limit based on the mean-free-path of the electrons and thus it is actually the smaller clusters in the cluster size distribution that are causing the highest energy acceleration?

We can confirm the conjecture of the reviewer. According to our simulations, larger clusters provide higher energy electrons. As a rough estimate, the optimal gain is realized for a half cycle transit. The associated kinetic energy scales as $(R_{\text{res}}/\omega)^2$, where R_{res} is the resonant radius. In the simulation we checked also cluster sizes of 2500, 5000 and 10000 atoms and find a similar increase. Note that the resonant radius is also a function of inner ionization, such that the ionization degree also has to be kept in mind. A fundamental requirement for the forward acceleration is that electrons can traverse the cluster without too

strong attenuation via elastic and inelastic electron ion scattering. Here, besides the lower density due to the expansion, the formation of a deep space charge potential suppresses collisions as it ensures high transit velocities with low scattering cross sections.

changes to the manuscript:

- we added a comment about the qualitative size-dependence of the acceleration to the main text: *“We note that the most energetic electrons in the experiment are expected to stem from the larger clusters in the size distribution. Simulations with various cluster sizes confirm this trend.”*
- we added a statement regarding the energy scaling to the discussion of the nanoaccelerator model: *“Here $v_{hct}=2R\omega\pi$ is the half cycle transit velocity, which also defines the characteristic energy scaling of the forward acceleration process with cluster size and laser frequency $E_{hct} \sim R^2 \omega^2$.”*

2) What was the initial state of the MD-silver clusters? The manuscript states the system was initialized with all Ag as +1 in an fcc. Were the electrons initialized with a total energy equal to their ionization potential or some global measure of the electron plasma temperature? Additionally, was the system simulated for both the pump probe or just the probe?

We agree that more information is useful for the reader.

changes to the manuscript:

- we added corresponding statements to the respective part of the methods: *“The conduction electrons were initialized with 3 eV kinetic energy, which roughly reflects the average valence electron kinetic energy in the bulk given by $3/5E_F$, where $E_F=5.5$ eV is the Fermi energy of silver. The simulations cover the full dynamics resulting from the excitation with pump and probe pulses.”*

3) What was the average final charge state of the silver clusters after irradiation? The main reason for this is to get an idea for the total number of electrons released with high energy and what proportion that represents to the total number of ionized electrons.

changes to the manuscript:

- we added the requested information to the MD section of the Methods : *“Within the resonant pump-probe excitation on average 8.2 additional electrons were removed from each atom, leading to a total inner ionization of $q_{ii}=9.2$ per atom. The electron emission into the continuum almost exclusively takes place in the probe pulse, reaching a total outer ionization degree of $q_{oi}=3.1$ for the considered parameters.”*

4) Was the full spatial extent of the laser pulse’s used or where the pulses “clipped”, removing the lower intensity wings of the pulse? This would help to explain if the low intensity signal is noise and it is just the high intensity region that provides the fast electrons or if it is some region of intensity that has the optimal parameters.

In the experiment no clipping was used. The presented data reflects the reaction products from the complete laser focus.

Typos

The paper is well written, but there are multiple mistakes of s → z that are incorrectly spelled: ionization, characterized and polarization.

We apologize – we checked the spelling again according to the guidelines (“Nature Communications uses Oxford English spelling.”) and corrected the corresponding typos.

REVIEWERS' COMMENTS:

Reviewer #1 (Remarks to the Author):

The authors have fully addressed all the concerns raised in my report. I recommend accepting the paper for publication in Nature Communication.

Reviewer #2 (Remarks to the Author):

I have only positive comments remaining for the article and look forward to sharing the results with my colleagues. I fully support the manuscript being published in Nature Communications.

Two typos I noticed in Data availability:
simulation code are availably on request
should be
simulation code are available on request

Within the resonant pump-probe excitation on
average 8.2 additional electrons where removed from each
atom
should be
Within the resonant pump-probe excitation on
average 8.2 additional electrons were removed from each
atom